# Pressure-Driven Perfusion System to Control, Multiplex and Recirculate Cell Culture Medium for Organs-on-Chips

**DOI:** 10.3390/mi13081359

**Published:** 2022-08-20

**Authors:** Mees N. S. de Graaf, Aisen Vivas, Andries D. van der Meer, Christine L. Mummery, Valeria V. Orlova

**Affiliations:** 1Department of Anatomy and Embryology, Leiden University Medical Center, Einthovenweg 20, 2333 ZC Leiden, The Netherlands; 2Applied Stem Cell Technologies, Technical Medical Centre, University of Twente, 7500 AE Enschede, The Netherlands; 3BIOS Lab on a Chip Group, MESA+ Center for Nanotechnology, Technical Medical Centre, Max Planck Institute for Complex Fluid Dynamics, University of Twente, 7500 AE Enschede, The Netherlands

**Keywords:** organ-on-a-chip (OoC), vessels-on-chip (VoC), fluidic circuit board (FCB), multiplexing, perfusion, recirculation

## Abstract

Organ-on-chip (OoC) devices are increasingly used to mimic the tissue microenvironment of cells in intact organs. This includes microchannels to mimic, for example, fluidic flow through blood vessels. Present methods for controlling microfluidic flow in these systems rely on gravity, rocker systems or external pressure pumps. For many purposes, pressure pumps give the most consistent flow profiles, but they are not well-suited for high throughput as might be required for testing drug responses. Here, we describe a method which allows for multiplexing of microfluidic channels in OoC devices plus the accompanying custom software necessary to run the system. Moreover, we show the approach is also suitable for recirculation of culture medium, an essential cost consideration when expensive culture reagents are used and are not “spent” through uptake by the cells during transient unidirectional flow.

## 1. Introduction

Microphysiological systems or “organ-on-chip” devices (OoCs) are an emerging class of pre-clinical models intended for modelling the physiology of healthy and diseased tissue and accelerating drug discovery [1]. OoCs are designed to mimic tissue microenvironments by combining different cell types in heterotypic cultures with (3D) extracellular matrix or synthetic polymer scaffolds, local chemical gradients and biomechanical stimuli that promote cellular responses in vitro to resemble those in vivo [2,3]. These complex cell culture devices are often intended for use with (micro)fluidic flow to mimic specific microenvironmental conditions. However, the throughput of experiments with controlled fluid perfusion in OoCs is still limited due to their technical complexity [4,5].

There are several reasons that fluidic flow is needed in OoCs [6]. First and foremost is the need for continual refreshment of the cell culture medium. Due to the small chamber and channel volumes in OoCs and often the high metabolic activity of cells under study, it is important that nutrients are replenished locally and metabolic waste products from cells are removed.

Secondly, fluidic flow can be used to introduce important mechanical stimuli. For instance, wall shear stress (WSS) is an important factor in determining how blood vessels react to stimuli in vivo and it is therefore important to include ways to measure and modulate WSS where biologically relevant [7].

Fluidic requirements depend on the type of OoC and the physiological condition of interest. Vessels-on-chip (VoC) generally require a large range in flow rates to induce both physiological and pathophysiological WSS depending on the model used [7,8]. In addition, complex cell culture medium formulations used for OoCs contain expensive components (e.g., growth factors). Recirculation of culture medium reduces the costs of experiments without compromising cell quality since culture medium supplements are in excess and are not “spent” in just one passage of the medium over the cells. Long-term perfusion would require excessive large volumes of cell culture medium. Furthermore, high medium volumes may dilute signaling molecules secreted by cells in complex culture systems to the extent that they may be beyond cellular or analytical detection. 

Different methods have been used to control perfusion in OoCs [9]. Each has its own advantages and disadvantages with varying degrees of required expertise before the technology can be implemented for OoC cell culture. When refreshing of the culture medium is the only requirement, gravity-driven flow is simple and effective. By applying a pressure head (P_head_) to the samples, the culture medium can be replaced without the use of external pumping systems. Alternatively, by placing the OoCs on a tilting rocker platform, bidirectional flow can be induced, and thus continuous refreshment achieved, again without the need for external pumping systems [10,11]. However, this approach is not suitable for all purposes as the flow rate is uncontrolled, “stop-and-go” and bidirectionality are of limited physiological relevance but can nevertheless influence the responses of the cells lining the channel [12,13]. Unidirectional flow can be achieved using gravity driven flow by designing more complex fluidic circuits; however, continuous and controlled flow is not realizable using gravity alone [14].

When it comes to controlled unidirectional flow, syringe pumps are the first choice as these are readily available and easily set up [15]. However, when used with compliant devices and flexible tubing, flow rate control in short-term experiments becomes less accurate than generally required due to long flow-settling times [16]. More importantly, the limited volume of syringes hinders their application in WSS experiments in long-term experiments due to the required high flow rates. Peristaltic or solenoid pumps are a better solution for long-term WSS experiments since they enable recirculation by continuously pumping fluids from and towards the same reservoir. However, the induced flow has a pulsatile character, especially at lower flow rates. This can be useful when modelling, e.g., pulsating blood vessels. Nonetheless, erratic and unintended mechanical (wall) stress can derive from such flow pattern [9,17,18]. Furthermore, the pulsating frequency is directly coupled to the flow rate, which limits flexibility and chip design when used for OoC-models.

To increase the number of samples per fluidic pump, OoCs can be placed in series or in parallel fluidic circuits. A caveat of these setups, however, is that cell debris and clumps can occur during experiments and (depending on channel dimensions) these can block the microfluidic channels or change flow distribution. When placed in series, for example, these blocks increase the internal pressure, which may be detrimental to cells and lead to experimental failure. In parallel circuits, the cell blocks can affect the mechanical stress of other samples in the circuit: the overall flow rate remains constant, but the flow rates for all other samples increase, resulting in differences in mechanical stress in each channel.

A major limitation of the positive displacement pumps described so far is their inability to run multiple compliant samples simultaneously. These systems can only control the overall flow rate in a system and lack control of mechanical and fluidic stress exerted on the individual OoC. Due to mechanical slack and system compliance reaching steady state flow, internal pressure may take up to minutes at the OoC [16,19,20]. As a result, positive displacement pumps lack control of flow distribution in compliant multiplexed fluidic circuits.

To control perfusion of complex fluidic circuits containing OoCs, pressure driven pumps are a better solution as the mechanical stress and perfusion parameters can be directly controlled. In the event of microfluidic blockades, perfusion conditions of the remaining samples do not change, protecting the remaining samples. Furthermore, compliance is better controlled as the internal pressure can be directly controlled and is not a result of all parameters being modified.

Here we developed a Python-based proportional–integral–derivative (PID) software controller able to integrate constant pressure differences into a fluidic circuit board (FCB), allowing accurate multiplexing of perfusion experiments. We demonstrate that the system can precisely control pressure driven flow rates in various microfluidic devices and recirculate culture medium without the need for more complex systems. The perfusion system is primarily designed for 3D VoC. As it can control both the flow rate and the internal pressure precisely, it protects sensitive cell cultures. At the same time, the system is useful and can be easily implemented other OoC systems that require accurate control of the flow rate and the internal pressure.

## 2. Experimental

### 2.1. Theoretical Model

The fluidic perfusion system was designed to apply a constant fluidic pressure difference (dP) on OoCs using a single input/output configuration (Figure 1a). The system design is composed of two pneumatic pressure controllers, a flow rate sensor and a micro-control unit (MCU) that enabled data streaming via serial communication. The pressure readouts are used to control the pressure controllers via custom-written software.

The system was able to control flow by controlling the air pressure difference of the reservoirs. The cell culture medium is then forced through the microfluidic circuit and collected in the opposite reservoir. By using the readouts from the pressure sensors, individual flow rates can be estimated using electrical circuit analogy according to Hagen–Poiseuille’s law as described by Oh et al. [21] Hagen–Poiseuille’s law describes the volumetric flow rate for a steady-state pressure-driven fluid flow:(1)$$\Delta P=Q\;R_{h}$$
where Δ*P* is the required pressure difference [Pa], *Q* is the volumetric flow rate [m^3^ s^−1^] and *R_h_* is the hydraulic resistance [Pa m^3^ s^−1^]

The hydraulic resistance of a microfluidic channel can be calculated with the following formulas.

For a rectangular channel:(2)$$R_{h}^{rec}\approx 12\mu \frac{L}{wh^{3}\left(1-0.63\frac{w}{h}\right)}$$
where *µ* is the dynamic viscosity [Pa s], *L* is the channel length [m], *w* is the channel width [m] and *h* is the channel height [m]. For a circular tube:(3)$$R_{h}^{cir}=128\mu \frac{L}{\pi d^{4}}$$
where *µ* is the dynamic viscosity [Pa s], *L* is the channel length [m] and *d* is the diameter [m]. 

Using these general equations and electrical circuit analogy, the flow distribution towards individual channels can be predicted for complex fluidic circuits using parallel and series circuits.

Total resistance for serial connected resistance:(4)$$R_{serie}^{total}=R_{serie}^{1}+R_{serie}^{2}+R_{serie}^{3}+\ldots +R_{serie}^{n}$$
where *R_serial_* are the individual fluidic components such as tubing, flow rate-sensors, microfluidic channels and OoC.

Total resistance for parallel connected resistance:(5)$$R_{parallel}^{total}={\left(\frac{1}{R_{parallel}^{1}}+\frac{1}{R_{parallel}^{2}}+\ldots +\frac{1}{R_{parallel}^{n}}\right)}^{-1}$$
where *R_parallel_* are the individual branches. Flow rate of individual samples using an FCB can then be calculated using:(6)$$Q_{n}=\frac{\Delta P_{OoC}}{R_{h}^{n}}$$
where *Q_n_* is the individual flow rate, $R_{h}^{n}$ is the individual hydraulic resistances and Δ*P_OoC_* is the controlled pressure difference across the samples. All calculations were performed, assuming cell culture medium containing 2% serum at 37 °C with a dynamic viscosity of 0.00079 [Pa s] [22].

### 2.2. Fluidic Circuit Board

To multiplex perfusion of the OoC devices, we used a fluidic circuit board (FCB) where multiple OoCs can be connected simultaneously (Figure 1b). Briefly, the fluidic circuit consists of a main feeder channel (Figure 1a, FCB green channel) that has individual branches towards the OoCs (yellow arrows). The fluidic flow is then collected in a central waste channel (red channel) that directs the flow towards the opposite, medium reservoir. In the central feeder and waste channels, two pressure sensors are inserted that act as the process variable for the PID controller run by the software. The FCB used can connect up to four OoC devices, each containing three microfluidic channels (Figure 1b).

### 2.3. Controlled Perfusion and Recirculation

Custom PID-controller software was written in Python to maintain constant pressure difference (dP) and internal sample pressure (*P_OoC_*) (Appendix A). The software measures two pressure sensors located at the inlet and outlet and corrects the pressure commands of the pressure controllers connected to the medium reservoirs. To increase the *P_OoC_*_,_ the back pressure of the receiving reservoir can be increased. By increasing the dP between the inlet and outlet, the flow rate can be expected to change in a linear fashion (Equation (6)). The software can control two different recirculating circuits: (1) an actively controlled valve-recirculation (Appendix A) or (2) a passively controlled microfluidic Graetz-recirculation circuit (Figure 1). The actively controlled recirculation requires the addition of two 3 port/2 position (3/2)-valves and a switchboard (Fluigent). By changing both the pressure commands and the position of these valves, the fluidic flow remains unidirectional in the OoCs [23]. The Graetz-recirculation circuit is a direct analogue of an AC/DC converter used in electrical engineering and requires 4 check valves in a similar orientation (Figure 1). This allows maintenance of unidirectional flow with only switching of the pressure commands, in contrast to method 1, where the actively controlled valves need accurate synchronization of both the pressure switch and valve positions to prevent backflow and pressure surges.

## 3. Materials and Methods

### 3.1. Fabrication of the Fluidic Circuit Board

The FCB was fabricated as described previously [23]. Briefly, it was composed of two cast 10 mm and one 15 mm polymethylmethacrylate Poly (methyl methacrylate) (PMMA) plates (Altuglass, Saint-Avold, France), where all connecting channels and fittings for Luer-slip connectors were milled with a CNC micro mill (Datron Neo, Datron AG, Mühltal, Germany). The dimensions of fluidic circuit are shown in Appendix A. After milling, the FCB was assembled as follows: both layers of the FCB were thoroughly cleaned using industrial cleaning wipes (Adolf Würth GmbH & Co, Duisburg, Germany), deionized water, absolute ethanol (Sigma-Aldrich, St. Louis, MO, USA) and propanol (Sigma-Aldrich). A solution of acetone (Sigma-Aldrich) in absolute ethanol at a volume ratio of 1:10 was added on top of the connection layer slab and the complementary channel layer slab was then pressed onto the connection layer slab and aligned using alignment pins (DIN 7-ISO 2338). The assembled FCB was then pressed at 1 kN at 55 °C using a hydraulic press (model 3889, Carver Inc., Miami, FL, USA).

### 3.2. Fabrication of Microfluidic Devices

The microfluidic OoC devices were fabricated from polydimethylsiloxane (PDMS, Sylgard 184, Dow Corning, Midland, TX, USA) using injection moulding as previously described [23]. The dimensions of the flow channels were as described previously (1.1 cm × 500 µm × 500 µm; l × w × h). Briefly, PDMS and base agent (Sylgard 184, Dow Corning) was mixed 10:1 (*w:w*) with curing agent and degassed. The degassed PDMS was transferred to a syringe and further degassed. The injection mould was assembled using six neodymium block magnets (N42, 1.3 T, approximately 60 N per magnet, Webcraft GmbH, Gottmadingen, Germany) and the PDMS was slowly injected. The filled injection-mould was set vertically at room temperature (19–22 °C) overnight. Afterwards, the PDMS was further cured at 75 °C for 60 min. The PDMS was carefully peeled off and excess PDMS was removed using a surgical knife. PDMS devices and round cover glasses (#1.5, ⌀30 mm Thermo Scientific, Waltham, MA, USA) were surface-activated using air plasma (45 s, 50 Watt at 60 Pa, CUTE-Femto Science, Hwaseong-si, Gyeonggi-do, Korea) and contact-bonded using light pressure.

### 3.3. Assembly of the Perfusion System

Fluidic and electronic components were obtained from commercial sources. The complete list of components used for the fluidic circuits described is shown in Table 1. Various fluidic circuits were used to demonstrate the different aspects of the setup and software and are shown at the relevant figure.

### 3.4. Pressure Controllers

Two EZ-LineUP (Fluigent, 345 mbar) pressure controllers were used to control the air pressure of the medium reservoirs. To connect the controllers to the laptop, the link-up module (Fluigent) was used. The left-most EZ-LineUP controller was designated as EZ1 and the right-controller as EZ2.

### 3.5. Flow Sensors

A thermal flow sensor (Fluigent) was connected to EZ1-LineUP. Two different ranges were used: size L (−1100 to +1000 µL/min) and size XL (−5500 to +5500 µL/min). The flow rate sensor of EZ1 measures the total flow rate of the circuit to validate perfusion. When experiments required a second flow sensor, this was connected to EZ2 and recorded using All-in-one software (A-i-O, Fluigent).

### 3.6. Pressure Sensors

Pressure sensors (MPRLS0300Y, Honeywell, Charlotte, NC, USA) were obtained and soldered onto a custom PCB which allowed the connection of ribbon wires according to the manufacturer’s instructions. These ribbon wires were connected to an MCU. The MCU was then connected to the controlling laptop via a USB. The MCU emits at 100 Hz 24-bit pressure output of 8 sensors. Alternatively, an Arduino MCU can be used (Baudrate:250000) To be able to read the MCU using the software, the serial port needs to be set to COM3.

### 3.7. Flow Rate Sensor Validation

A flow rate sensor (Fluigent, XL) was thoroughly cleaned by rinsing with 70%/30% ethanol/ddH_2_O, followed by incubation with hypochlorite (Glorix, Unilever, London, UK) for 20 min. After incubation the sensor was subsequently rinsed with 10 mL ddH_2_O, 10 mL propanol and dried with nitrogen. The flow sensor was used in a fluidic circuit connected to a 10 mL pipette (Figure 2). Endothelial Cell Growth Medium -2 (EGM-2) containing 2% serum (in anticipation of future of use for endothelial cells in vessel-on-chip models) was used as fluid and the Flow-EZ was set to fixed flow rate using the manufacturer’s PID-loop at room temperature. At every millilitre mark, the time-delta was noted for at least 3 mL. Using this time-delta, the flow rate was calculated and compared to the set flow rate. The setup was allowed to recirculate for 48 h and the experiment then repeated.

### 3.8. Liquid Level Measurement

Standard 15 mL Falcon tubes combined with 4-port pressure caps (Elveflow) were used as medium reservoirs. Two different methods were used to insert the required pressure sensors.

To directly measure the liquid level, a hole was punched using a 2 mm biopsy puncher in the bottom of the 15 mL Falcon tube to create a leak-tight seal with the long-port of the sensor (Figure 3a). Care was taken to remove any air pockets between the sensor and the liquid.

To remotely measure the liquid level, a tube was placed at the low liquid level through one of the ports of the pressure cap and the reference sensor was inserted (Figure 3b).

### 3.9. Graetz Recirculation Circuit Validation

Two flow sensors (size L) were first calibrated against each other and placed in the fluidic-circuit as shown in Figure 4. Using the software, the “recirculation algorithm” was executed and measured using the A-i-O Fluigent at 20 Hz.

### 3.10. Validation of Flow Rate Control Using Applied dP

Two microfluidic devices with a total of 6 channels were connected to the FCB. The other ports were blocked using dummy chips (no perfusable channels). The circuit was primed using equilibrated EGM-2 at 37 °C. The medium was equilibrated by placing a culture flask of medium inside an incubator (37 °C, 5% CO_2_) for at least 2 h. Fluidic reservoirs were placed into a heat block set at 37 °C. A flow rate sensor with a maximum of 1100 µL/min was used (Fluigent, size L). The system was completely bled from air and pressure sensors were correctly calibrated. The dP was allowed to increase slowly until the maximum of the flow sensor was reached; the flow rate was then plotted against the measured pressure difference. Measurements were performed on a single channel, while all other channels were blocked using dummy chips, and the experiment was repeated with 6 channels connected to the FCB.

### 3.11. Long-Term Perfusion

An 8 cm length of tubing (diameter: 250 µm) was placed between two T-junctions. The pressure sensors were inserted into the available port and the system was primed with ddH_2_O. All sensors were correctly calibrated, and the system was allowed to recirculate for 48 h with a dP of 29 mbar.

### 3.12. Code Availability

The Python code can be found at the GitHub repository [24].

## 4. Results and Discussion

### 4.1. Design Perfusion Platform

To overcome current limitations of OoC-perfusion, we developed a perfusion system that controls all fluidic parameters exerted on the sample. This developed system controls the perfusion rate and mechanical stress on the cells of multiple OoCs, achieves recirculation without the use of actively controlled valves and measures liquid levels to eliminate the risk of emptying medium reservoirs.

The system consists of two pressure controllers, additional pressure sensors and custom written controlling software (Figure 1 and Appendix A). The system is designed to control multiplexing flow in using an FCB by maintaining a constant pressure difference (dP). However, the system can also be used for the perfusion of a single OoC (Appendix A) or to multiplex the perfusion using manifold splitters (Appendix A).

The custom-written software is a Python-based, PID controller that integrates pressure sensors placed at the OoCs to stabilize fluidic flow (Appendix A). Furthermore, the custom software controls the recirculation by switching commands to prevent the medium reservoirs from running empty. The software registers all sensor data for post experiment analyses. After each direction switch, the software summarizes all data for that sequence and sends the data to a desired email account for online monitoring of the experiment.

### 4.2. Characterization of Pressure Controllers and Flow Rate Sensors

To control the perfusion of the OoCs, two Flow-EZ™ (Fluigent) were used. Flow-EZ ™ is a pressure controller that is used to accurately control the air pressure of a medium reservoir to drive fluids through the microfluidic circuits. These pressure controllers have two controlling modes: a constant pressure mode and a constant flow rate mode. The constant flow rate mode can be used as alternative to a positive displacement pump; however, the maximum applied pressure can be limited. The constant pressure mode can be used to apply a constant force. Both modes use an internal PID controller to transmit the process variable from either the internal pressure sensor or an external flow rate sensor integrated in the fluidic circuit.

The Fluigent-compatible flow rate sensor is thermal mass-based and commonly used for high accuracy lab-on-a-chip purposes. These sensors apply and measure a temperature gradient to calculate the mass flow. To investigate the suitability of these sensors for OoCs in long-term experiments, we examined the accuracy of a clean sensor using cell culture medium and re-measured the same sensor 2 days after continuous perfusion (Figure 2). This showed that for short-term experiments, this sensor is accurate in measuring flow rates in cell culture medium (EGM-2) but that its accuracy is highly sensitive to fouling by culture medium when used over longer periods. When a flow rate sensor becomes fouled, it significantly underestimates the true flow rate (Figure 2b). This inaccuracy resulting from fouling has important implications when fixed flow rate PID-mode is used for long-term experiments. The PID-loop enforces higher pressures than required and the real flow rate can quadruple (Figure 2b, blue plot) at the higher ranges of the sensor. This leads to exposure of the OoCs to higher mechanical stress than anticipated and possible misinterpretation of the perfusion experiments.

We further investigated the functionality of the constant pressure mode as we opted to perfuse parallel circuits using constant pressure. We found that the flow rate decreases when a constant pressure is applied to the reservoir liquid level (Figure 5). Conversely, when using the constant flow rate mode, the required pressure is increased. This observed effect is due to the P_head_ difference when the fluid is being transferred from one reservoir (decreasing positive contribution) to another (increasing resistance) (Figure 5a). Approximately 1.2 to 1.5 mbar per displaced ml of fluid was observed depending on fluidic density and used reservoir (Appendix A). Depending on the total resistance of the circuit, this can lead to a significant decrease in flow rate (Figure 5a). This effect can be minimized by applying higher hydraulic resistance in front the OoCs; however, this solution may lead to undesired increased residence time or shear rates inside the tubing.

### 4.3. Liquid Level Measurement

The inaccuracy of the flow sensor following long-term exposure to cell culture medium also affects experiments in recirculation systems. To determine the correct timing for switching, the output of the flow sensor is used to calculate the total displaced volume. When this measured value is lower than the actual flow rate, medium reservoirs eventually empty and experiments fail [23]. To control the flow rate and recirculation without relying on these flow rate sensors, we integrated additional pressure sensors at the OoC and medium reservoirs.

Because of the P_head_ of the liquid, the medium level can be determined by measuring the pressure difference between the bottom of the reservoir and the air pressure above the liquid level. This can be achieved by inserting pressure sensors at the bottom of the reservoir (Figure 3a). This method is highly accurate, but due to direct exposure of the sensor to cell culture medium, the sensor may become fouled, less accurate and a source of (bacterial) contamination. Therefore, we investigated if we can accurately measure it remotely via a tube (Figure 3b). Although indeed accurate, we observed that when a reservoir is pressurized, the liquid is forced into the measurement tube and an abrupt increase in pressure is measured. When the pressure is reduced, the measured fluid decreases again. We tested two tube-diameters (0.8 mm and 3 mm) and found that by using a large tube diameter this measured effect is minimized.

### 4.4. Recirculation Using the Graetz-Recirculation Circuit

OoCs require relatively high flow rates to mimic many physiologically relevant conditions. However, if there are relatively few cells in the OoC, important signaling molecules may be diluted beyond detection by (other) cells or sensors. As experiments may take several weeks, it is important to be able to recirculate cell culture medium for cost effectiveness. Pressure driven fluidic circuits require actively controlled valves that need to be synchronized with the pressure commands [23]. These valves are often bulky, challenging to implement and a potential source of bacterial contamination. Here we developed passive recirculation circuit based on an AC/DC converter circuit. It required four check valves oriented in a similar fashion as the diodes in a Graetz bridge (Figure 4a). This circuit is a cost effective, small footprint method to achieve unidirectional recirculation by simply switching the pressure of the pressure controllers (Figure 4b). To minimize disruption, it is essential to match the PID parameters to the fluidic circuit which can be easily done within the software we developed (Figure 4a,b). We also observed the importance of quickly venting overpressure as this affects the settling time (Figure 4b,c). To validate the flow rate in the OoCs, we added fluorescent beads to the medium and recorded bead displacement during each directional switch (Appendix A). This showed minimal disruption in the flow rate. The check valves we used are disposable, which reduces the risk of bacterial contamination by inadequate cleaning or rinsing. Furthermore, the passive fluidic circuit is less complex to implement and easier to scale as it requires no electronic wires or software. Microfabrication techniques have been described that further miniaturize the check valves in the microfluidic chips, minimizing swept and dead volumes [25]. For practical reasons, the inlet and outlet bridges are separated for long-term experiments as this allows the separation of inflow and outflow (Figure 1a).

To control the flow rate of the OoCs, we applied a constant dP across the samples using the custom software. To validate this approach, we set up an FCB with six samples and an FCB with one sample (Figure 6). We compared these with the predicted values (Appendix A). Both plots showed the linear relation between the dP and flow rate, and the single sample correlated with the predicted flow rates. However, the multiplexed sample showed deviation from the model. These differences could be attributable to fabrication variation of the microfluidic devices used and the FCB.

To test the final system for long-term perfusion, we set up a single chip and perfused it for 48 h at a target pressure difference of 29 mbar to reach flow rate of approximately 200 µL/min with ddH_2_O to validate how the algorithm maintains an equal flow rate for over 48 h (Figure 7c). These results were as expected, validating this system for further use.

## 5. Current Limitations of the Presented System

Despite the advance the system we describe represents, there are some limitations that remain. The controlling software takes input from the 2 separate gauge pressure sensors connected to the fluidic board. Preferably this would be done using a differential sensor but at present, wet/wet differential sensors with the desired resolution were not available in a miniaturized format that would fit this concept design. The used MicroPressure sensors (MPR, Honeywell) have a small footprint, are low cost and can be used in both wet and dry conditions. By dampening the signal of 2 separate gauge pressure sensors, the output could be used to calculate the dP. This approach resulted in stable and repeatable flow control despite the limitations of using two separate gauge pressure sensors for differential measurements as shown in this work, although this does affect the response time of the system.

## 6. Conclusions

In this work we investigated and optimized the use of pressure-driven flow circuits for OoC perfusion. We demonstrated that thermal flow rate sensors are not suitable for long-term cell cultures as these would affect flow rates and recirculation algorithms. We therefore optimized the OoC systems by stabilizing the flow rate using pressure sensors as they can control the flow rate in a more stable way and protect cells from unwanted mechanical stress. By measuring the liquid levels using pressure sensors, a reliable and flexible recirculation algorithm was developed to perfuse for extended periods of time. We fabricated a passively controlled recirculation circuit from off-the-shelf components, which is an economical and less complex solution to other (commercially available) solutions. The software controlling the perfusion system is intuitive and can be used for different microfluidic devices at different flow rates without modification.

## Figures and Tables

**Figure 1 micromachines-13-01359-f001:**
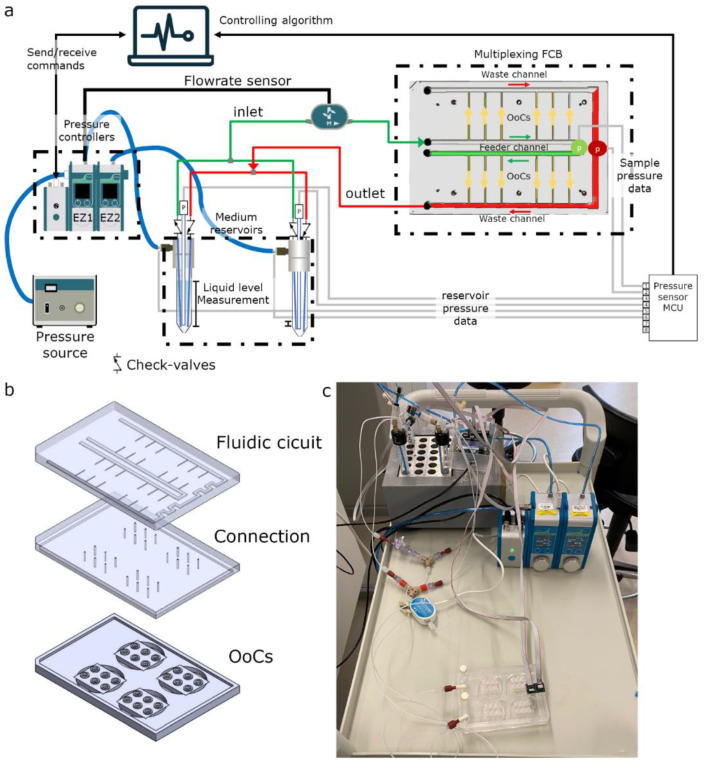
Overview of the perfusion system (**a**) Schematic of the microfluidic circuit consisting of 2 pressure controllers connected to a pressure source, additional pressure sensors (p), a passively controlled recirculation circuit and a custom-made fluidic circuit board to multiplex the perfusion to multiple samples. (**b**) Exploded view of the FCB shows the fluidic circuit, connection layer and 4 OoC-devices. (**c**) Photograph of the system tested.

**Figure 2 micromachines-13-01359-f002:**
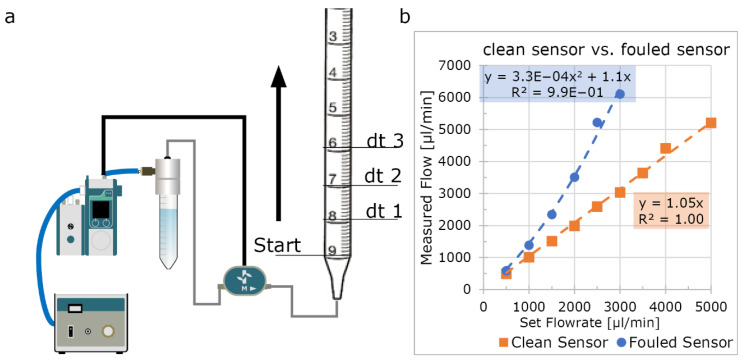
Validating flow rate sensor accuracy. (**a**) Schematic of the fluidic circuit used to validate sensor accuracy. A 10 mL serologic pipette was attached to the outlet of the circuit, and the pressure controller was set to a fixed flow rate using the internal PID-loop. At multiple intervals, time was noted and the flow rate was calculated (**b**) A clean sensor accurately measures the flow rate of cell culture medium (orange plot); after two days of continuous perfusion, the sensor becomes inaccurate (blue plot).

**Figure 3 micromachines-13-01359-f003:**
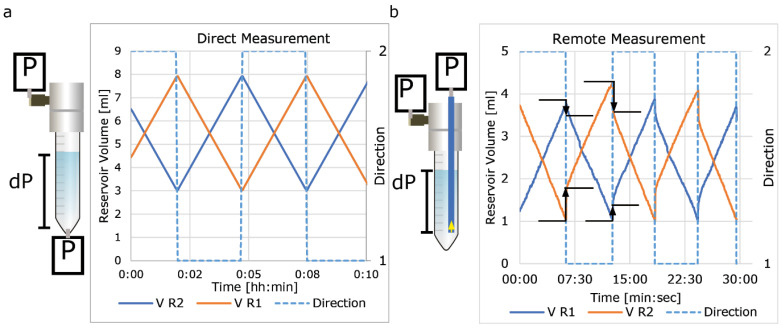
Measuring liquid level using pressure sensors. (**a**) Measurement of liquid level using direct measurement. (**b**) Remote measurement using tubes with 2 different diameters. Remote measurement is accurate; however, there is an offset when reservoirs are pressurized. This offset is diameter-dependent and larger with a small diameter (800 µm, orange) than with a larger diameter (3 mm, blue plot).

**Figure 4 micromachines-13-01359-f004:**
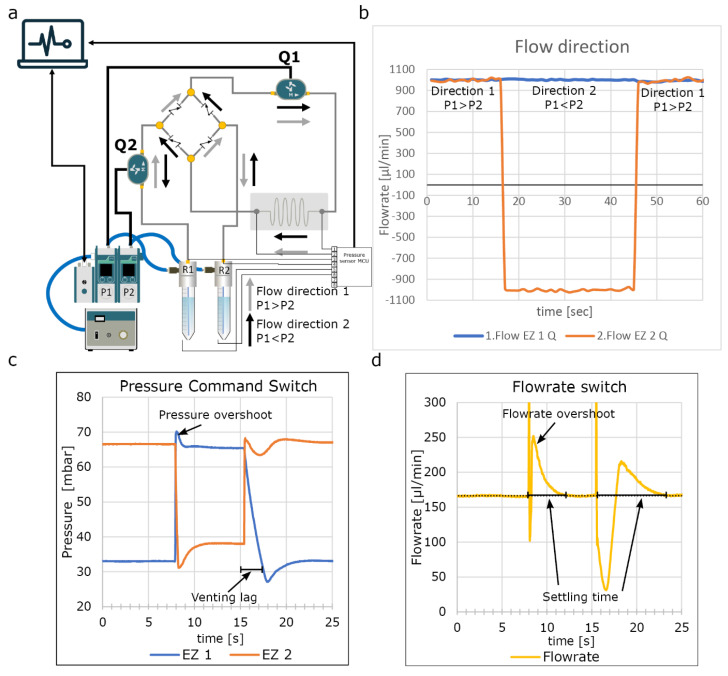
Graetz recirculation circuit. (**a**) Schematic of the fluidic circuit used shows the microfluidic analogy of the Graetz bridge used in an AC/DC convertor using check valves; if the pressure in R1 is higher than R2, fluidic flow follows the gray arrows. When the pressure is reversed, fluidic flow follows the black arrows, remaining unidirectional at Q1 and bidirectional a Q2. (**b**) Direct measurement of the recirculation. When pressure commands are reversed, the flow is reversed at flow rate sensor Q2, however, remains unidirectional at flow rate sensor Q1. (**c**) High resolution recording of the pressure switch at the fluidic reservoirs shows small overshoot and undershoot in pressure resulting in spike in flow rate. Interestingly EZ1 vents slower than EZ2 affecting the algorithm. (**d**) Measured flow rate shows the overshoot due to the pressure overshoot and settling time of approximately 4 s, due to the venting lag of EZ1 settling time is approximately 8 s. See Appendix A to see perfusion of suspended micro beads during switching.

**Figure 5 micromachines-13-01359-f005:**
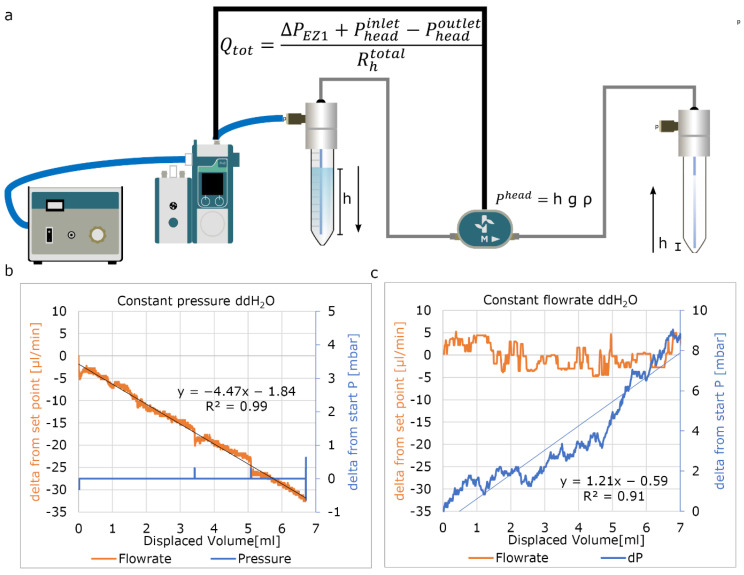
Controlling flow rate with fixed parameters. (**a**) Used fluidic circuit. (**b**) Applying a fixed pressure difference results in a constant decrease of the flow rate per displaced volume due to effect of P_head_. (**c**) Fixed flow rate shows a steady increase in applied pressure due to the changing P_head_ using fixed flow rate.

**Figure 6 micromachines-13-01359-f006:**
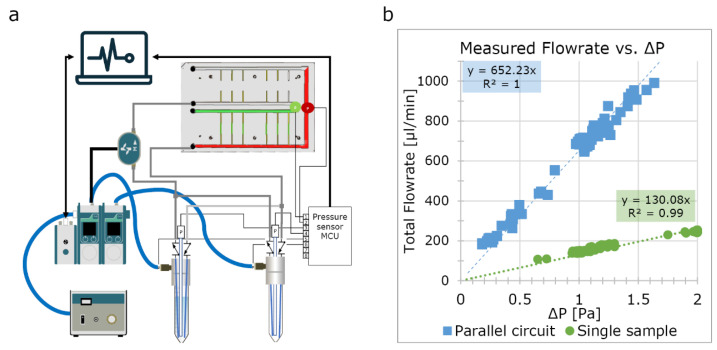
Flow rate vs. pressure difference. (**a**) Schematic of the used fluidic circuit shows the FCB with 6 channels connected. The free channels were blocked using a dummy chip. (**b**) Pressure difference relationship for a single channel (green plot) and six channels in parallel (blue plot). The measurements show correlation with the predicted values (Appendix A); however, the six channels in parallel show a 5-fold increase in flow rate dependence but this can be attributed to the tolerances of the fabricated FCB.

**Figure 7 micromachines-13-01359-f007:**
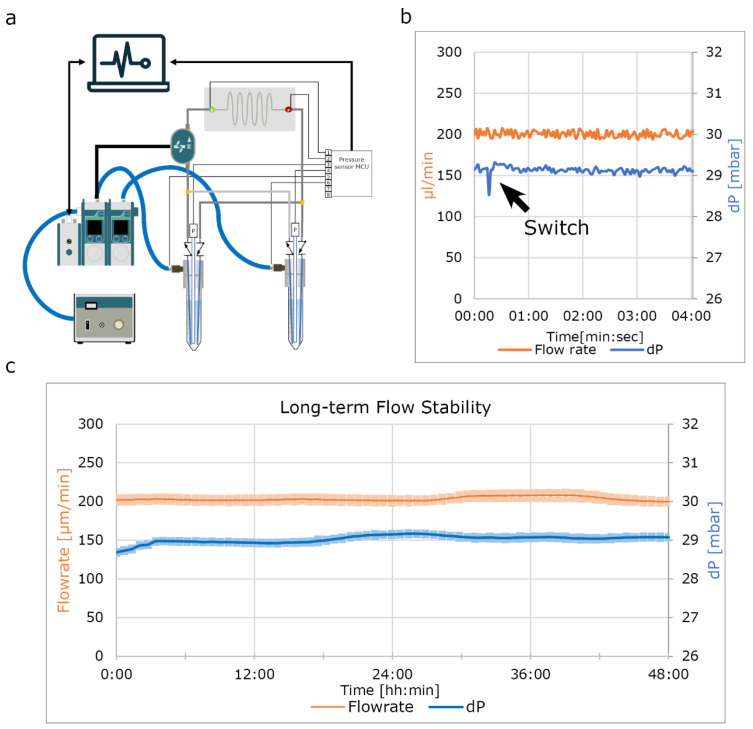
Long-term flow rate stability. (**a**) Schematic of the used fluidic circuit shows a single sample. For this experiment ddH_2_0 was used to demonstrate the functionality of the software in maintaining constant flow rate without the effect of sensor fouling. (**b**) Flow rate and pressure difference for 4 min during a flow reversal shows little variation. (**c**) Two day perfusion using ddH_2_O shows minimal variation. Error band shows the interquartile range of that time segment.

**Table 1 micromachines-13-01359-t001:** Materials used for assembly of various fluidic circuits.

Material	Qty.	Manufacturer	Supplier	Product SKU:
FCB or single microfluidic device	-	Custom	-	-
Pressure cap for 15 mL Falcon-S (4 port)	2	ELVEFLOW	Darwin-microfluidics	LVF-KPT-S-4
15 mL reservoirs	2	Falcon		
Check valves	4	Master flex	Darwin-microfluidics	MF-30505-92
Flow sensor L or XL *	1 or 2	Fluigent	Fluigent	FLU_L_D_FDG
Flow EZ-line Up 345 mbar *	2	Fluigent	Fluigent	LU-FEZ-345
Link-up	1	Fluigent	Fluigent	LU-LNK-0002
Pressure sensors *	4–6	Honeywell	Farnell	MPRLS0300YG0001B
Luer to 1/16 barb	24	IDEX	Darwin-microfluidics	CIL-P-854
¼-28–Female-to-male Luer adapter	2	IDEX	Darwin-microfluidics	CIL-P-655-01
3-way valve	2	IDEX		
¼-28–Female-to-Female Luer Lock adapter	2 in	IDEX	Darwin-microfluidics	CIL-P-678
Y-connectors	2	IDEX	Darwin-microfluidics	CIL-P-512
PTFE tubing–1/16” OD X 1/32” ID*		ELVEFLOW	Darwin-microfluidics	LVF-KTU-15
Printed circuit board (PCB) for pressure sensor	1	custom		
Arduino or otherMCU	1	Arduino		
Ribbon wires	4–6			
**optional:**				
Microfluidic Manifold 9 Port	2	ELVEFLOW	Darwin-microfluidics	LVF-KMM-02
2-Switches	2	Fluigent	Fluigent	2SW002
Pressure source 1.2 bar	1	Fluigent	Fluigent	FLPG005
* Pressure and flow-rate range are setup depended				

* This means that although we use here a 345 mbar controller, it may not be the best option for specific fluidic circuits and therefore the reader should be reminded that there are alternatives.

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
