# Peer review of "Pressure-Driven Perfusion System to Control, Multiplex and Recirculate Cell Culture Medium for Organs-on-Chips"

_micromachines, 2022, doi:10.3390/mi13081359_

Round 1

Reviewer 1 Report

Recommendation: Major revisions needed as noted.

 I have thoroughly reviewed the manuscript entitled "Pressure-driven perfusion system to control, multiplex and recirculate cell culture medium for Organs-on-a-Chip". This manuscript describes the development of controllable perfusion system for applying various microfluidic devices and recirculation of culture medium without the integration of complex systems.

Taking into account the quality of work and scope of the journal, I would recommend the major revision according to the following comments. Some suggestions were listed as follows for the authors to improving the work.

 Comments:

# 1. Although the research topic raised by author is interesting and crucial in the field of Organ-on-a-chip (OOC), I am afraid that the current version of the manuscript should be improved in the point of organ types (e.g., blood vessel, heart, lung, brain, etc.). The author should be clearly described the target organs for using this system and added the examples with relative references to mimic the physiological microenvironments in human.

# 2. I suggest the authors to demonstrate cell culture integrating this system for the clear understanding of the readers.

# 3. Why require the recirculation to operate OOC system? To mimic microphysiological system using OOC, media change is best way to supply nutrients and oxygen for the culture of cells. How to remove waste from cells in recirculating culture medium?

# 4. The author should check `author guideline` from this journal and modify the references. Current version is not appropriate.

Author Response

We thank the reviewer for critically reviewing our manuscript and providing constructive comments. We now have addressed all issues raised by the reviewer and provide a point-by-point response below, as well as a revised manuscript with highlighted changes. We feel that the manuscript has been improved significantly.

Reviewer 1:

I have thoroughly reviewed the manuscript entitled "Pressure-driven perfusion system to control, multiplex and recirculate cell culture medium for Organs-on-a-Chip". This manuscript describes the development of controllable perfusion system for applying various microfluidic devices and recirculation of culture medium without the integration of complex systems.

Taking into account the quality of work and scope of the journal, I would recommend the major revision according to the following comments. Some suggestions were listed as follows for the authors to improving the work.

Comments:

Point # 1. Although the research topic raised by author is interesting and crucial in the field of Organ-on-a-chip (OOC), I am afraid that the current version of the manuscript should be improved in the point of organ types (e.g., blood vessel, heart, lung, brain, etc.). The author should be clearly described the target organs for using this system and added the examples with relative references to mimic the physiological microenvironments in human

We agree that the target organ application was not made clear in the original version of the manuscript. We have elaborated on this in the revised version to ensure that the reader understands that the system is primarily designed to multiplex controlled perfusion and pressure in three-dimensional (3D) Vessels-on-Chip (3D VoC). However the need for perfusion is organ type independent. For this reason, no cell culture integration experiments are shown in this manuscript but we currently have a separate manuscript under review in the Lab on a Chip journal where we demonstrate the utility of the system for 3D VoC fabricated using viscous finger patterning methods, as previously described (de Graaf et al., 2019).

At the same time, we believe that the system is useful and can be easily implemented into any other OoC systems that require accurate control of the flow rate and internal pressure.

We, therefore, added a paragraph at the end of the introduction section to explain this(lines 99-103):

“The perfusion system is primarily designed for 3D VoC, as it can control both the flowrate and the internal pressure precisely, protecting sensitive cell cultures. At the same time, the system is useful and can be easily implemented other OoC systems that require accurate control of the flow rate and the internal pressure..”

# 2. I suggest the authors to demonstrate cell culture integrating this system for the clear understanding of the readers.

As mentioned above, the reason no cell culture integration experiments are shown in this manuscript is that we currently have a separate manuscript under review in the Lab on a Chip journal where we demonstrate the utility of the system for 3D VoC fabricated using viscous finger patterning methods, as previously described (de Graaf et al., 2019).This manuscript can be made available to the reviewer upon request.

# 3. Why require the recirculation to operate OOC system? To mimic microphysiological system using OOC, media change is best way to supply nutrients and oxygen for the culture of cells. How to remove waste from cells in recirculating culture medium? 

The main reason for choosing recirculation is the cost of the medium consumed in continuous perfusion experiments, which are common in 2D and 3D VoC culture, i.e. low to medium flow would require 20 to 100 µl of medium per minute (for typically used systems). Therefore, continuous perfusion over the period of 6 to 24 hours without recirculation will require 36 to 144 ml of medium per VoC. In addition, recirculation of small volumes is critical when the availability of primary material is limited, such as human leukocytes or human plasma from patients. Finally, recirculation is useful for analyzing the cytokines used or secreted by the cells to prevent the sample from diluting when high volumes are used. Removing waste from cells is not an issue as the medium-to-cell ratio is still relatively large compared to conventional cell culture experiments.

We added a paragraph to explain this reasoning (line 43-48):

“complex cell culture medium formulations used for OoCs contain expensive compo-nents (e.g. growth factors). Recirculation of culture medium, reduces costs of experi-ment without compromising cell quality since culture medium supplements are in ex-cess and are not "spent" in just one passage of medium over the cells. Long term perfu-sion would require large volumes of cell culture medium. Furthermore, high medium volumes may dilute signaling molecules secreted by cells in complex culture systems to the extent that they may be beyond cellular or analytical detection.”

# 4. The author should check `author guideline` from this journal and modify the references. Current version is not appropriate.

We have updated the references to the style of the journal.

Author Response

We thank the reviewer for critically reviewing our manuscript and providing constructive comments. We now have addressed all issues raised by the reviewer and provide a point-by-point response below, as well as a revised manuscript with highlighted changes. We feel that the manuscript has been improved significantly.

Reviewer 2:

Overview

This article described the use of a custom pressure driven flow system for the purpose of Organ on a chip media perfusion. Pressure sensors are used to stabilize the flow rate over long time periods and also reduce the mechanical stress experienced by cells in comparison to the other described systems. The authors also show that thermal-based flow sensors are not suitable for OoC media perfusion since they are not able to reliably calculate the flow rate of cell suspensions over time periods of days.

General comments

#1 The authors say that “erratic and unintended mechanical (wall) stress can derive from such flow pattern” (Line 62-63) when discussing the use of peristaltic flow perfusion systems. However, no references are provided, therefore this appears to be a statement/ opinion. Are they able to provide evidence (ie, provide references) that this type of infusion system does indeed cause unintended WSS?

We have now added a reference where the major pumping systems in microfluidics are reviewed. In this review, the unintended and erratic flow pulses are addressed and compared to more stable systems such as pressure driven flow. In the following reference the pulsatile character typical of peristaltic pumps is evidenced by the oscillatory boundary position of the two liquid interphases. [line 71].

https://analyticalsciencejournals.onlinelibrary.wiley.com/doi/epdf/10.1002/elps.201300205

#2 In the next paragraph (beginning Line 64), the authors provide a reasonable argument about the limitations of positive displacement perfusion systems, however, this is again not backed up by credible evidence. Can evidence please be identified in the literature which provides the reader with confidence in the given arguments? Otherwise, if this is based on experimental data, can this data be shown in perhaps the ESI? It would be a necessary control if no literature exists which verifies such claims.

We are referring here to the long startup time of positive displacement pumps when used with compliant samples.

These systems are able to accurately control output flow, however, when ramification of the channels occurs (or microbubbles are present), these systems are not able to ensure the correct flow distribution since this relies on the geometry of the ramification of the micro channels.
This effect has been also addressed in the medical field of anesthesiology.

We clarified the paragraph as follows and added relevant references [line 82-88]:

“A major limitation of the positive displacement pumps described so far is their inability to run multiple samples simultaneously. These systems can only control the overall flow rate and lack control of mechanical and fluidic stress exerted on the individual OoC. Due to mechanical slack and system compliance, reaching steady state flow and internal pressure may take up to several minutes at the OoC[16-18]. As a result, positive displacement pumps lack control of flow distribution in compliant multiplexed fluidic circuits.”

#3 Rn has a different meaning whether serial or parallel connected resistance, perhaps it is useful to use an another abbreviation in the second case so that is not confusing. For example, Rsn or Rcn.

We have included this suggestion as:

Rserial and Rparallel (Eq. 4 and Eq. 5)

#4 Fig S2a is called upon before Fig S1a – can the authors check and where necessary correct the order/ referencing of figures within the text?

We reordered the naming of the figures in the text [fig s1 line 155,fig s2 line161)

#5 Actual artwork of fluigent components is used in the graphics – are permissions required from fluigent to reproduce their content? It may be necessary to check with Micromachines and/or Fluigent.

The authors have permission from Fluigent to use their artwork. Fluigent endorses the use of their artwork which is available on their website for this purpose.

www.fluigent.com/resources-support/support-tools/downloads/fluigent-product-icons-images/fluigent-product-icons-images/
“Images for generating system schematics. For scientific and commercial use”

#6 Line 175 – did you really apply pressure during the initial contact when bonding the PDMS channels? This is usually described as not advisable in the literature, since an uneven pressure is typically applied over the surface of the channel. This can cause deformation of the PDMS and then result in a channel geometry which is not the same as the intended design.

The authors are aware of this type of deformation. However, light pressure was deemed necessary for optimal contact bonding due to the quality of the contact surface of the micro milled molds used in this study. Since the height of the channels used is relatively large (500 µm), the relative deformation is expected to be small. This is also stated in the manuscript (line 368-370):

“However, the multiplexed sample showed deviation from the model. These differences could be attributable to the fabrication variation of the microfluidic devices used and the FCB.”

Specific comments

Line 60: this should read “pumping” rather than “pump”

Thank you for raising this point; we have corrected it.

Line 105: the theoretical equation for calcuting viscosity is not given – please provide this. Also, which type of viscosity is this (dynamic, or kinematic)?

We clarified the sentence

“where µ is the dynamic viscosity [Pa s], L channel length[m], w channel width [m] h: channel height[m] and for a circular tube”

Line 145: are there 4 check valves in figure 2? If so, can you indicate these in the figure and/or legend? It is not obvious that 4 check valves exist within the circuit, nor where they are located.

We referred to the wrong figure as it should be figure 1. Now corrected.

Lines 146-147: the statement about synchronization does not completely make sense it is current format. Can this be revised?

We agree that the statement does not make sense as we did not elaborate enough on the subject. We revised the paragraph as follows: [line 160-170]

“The software can control two different recirculating circuits: (1) an actively controlled valve-recirculation (Fig. s2a) or (2) a passively controlled microfluidic Graetz-recirculation circuit (Fig. 1). The actively controlled recirculation requires the addition of two, 3 port/2 positions (3/2)-valves and switchboard (Fluigent). By changing both the pressure commands and the position of these valves, the fluidic flow remains unidirectional in the OoCs [19]. The Graetz-recirculation circuit is a direct analogue of an AC/DC converter used in electrical engineering and requires 4 check-valves in a simi-lar orientation (Fig. 1). This allows maintainance of unidirectional flow with only switching of the pressure commands, in contrast to method 1, where the actively controlled valves need accurate synchronization of both the pressure switch and valve positions to prevent backflow and pressure surges.”

Line 151: Where was the fabrication of the FCB previously described? No reference is provided.

We updated the reference.

Table S2: the viscosity of what? Presumably the media? Is this the dynamic viscosity?

Thank you for pointing this out. We updated Table S2 by explicitly stating it is the dynamic viscosity of a 2% serum-containing cell culture medium at 37°C.

Line 164-165: as previously described where? Please clarify.

We updated the reference.

Line 179 : The reference to “table 1” is not capitalized.

Thank you for noting this error; we corrected it.

Round 2

Reviewer 1 Report

The manuscript was corrected to the sufficient level for Micromachines